# Reuse of Water Contaminated by Microplastics, the Effectiveness of Filtration Processes: A Review

**Juan A. Conesa** [1,2,*] and **Nuria Ortuño** [1,2]

1 Department of Chemical Engineering, University of Alicante, P.O. Box 99, E-03080 Alicante, Spain; nuria.ortuno@ua.es
2 Institute of Chemical Process Engineering, University of Alicante, P.O. Box 99, E-03080 Alicante, Spain
* Correspondence: ja.conesa@ua.es; Tel.: +34-9-6590-3400

**Abstract:** Water treatment generally does not specifically address the removal of microplastics (MPs). Nevertheless, treatment plants process water effectively, and the number of synthetic microparticles in effluents is usually very low. Still, discharge volumes from water-treatment plants are often elevated (reaching around $10^8$ L/day), leading to the daily discharge of a substantial number of MPs and microfibers. Furthermore, MPs accumulate in the primary and secondary sludge, which in the end results in another environmental problem as they are currently used to amend soils, both for cultivation and forestry, leading to their dispersion. Something similar occurs with the treatment of water intended for human consumption, which has a much lower but still significant number of MPs. The amount of these pollutants being released into the environment depends on the processes that the water undergoes. One of the most-used treatment processes is rapid sand filtration, which is reviewed in this article. During the filtration process, MPs can break into smaller pieces, resulting in a greater number of plastic particles which mainly accumulate in sewage sludge. Thermal processes, such as incineration, carried out in facilities with the best available techniques in practice, could guarantee the safe disposal of highly MP-contaminated sewage sludges.

**Keywords:** drinking water; wastewater; sewage sludge; microplastics; microfibers; sand filtration

## 1. Introduction

The global demand for water is expected to increase by nearly one-third by 2050. Investing in water recovery and reuse could save up to 90% in energy and 70% in water usage, according to the United Nations (UN) [1]. Several of the Sustainable Development Goals (SDGs) are related to effective wastewater treatment. In objective 6, "Clean water and sanitation", one of its goals is to improve water quality by reducing pollution, eliminating discharge and minimizing the emission of chemical products and hazardous materials, increasing safe reuse worldwide. Likewise, objective 11, "Sustainable cities and communities", advocates for reducing the negative environmental impact per capita of cities, including paying special attention to air quality and the management of various toxic effluents.

Typical water-purification systems consist of pretreatment units (generally screening and clarification) and primary treatment that continues in a biological treatment unit (which constitutes secondary treatment). The primary treatment of wastewater consists of coagulation or flotation (for the removal of fats) followed by sedimentation. Afterwards, the organic matter in suspension is eliminated through a secondary treatment in the activated sludge reactor [2]. Many treatment systems also have tertiary units (often consisting of sand filtration or membrane-based filtration), ensuring a high-quality final effluent, whether for reuse in agriculture or safe disposal. Finally, a disinfection process is carried out (with chlorine, ozone or ultraviolet light) to eliminate pathogenic pollutants before discharging the treated water into the effluents. A simplified schematic representation of the processes that are generally present in a wastewater-treatment system is provided in Figure 1.

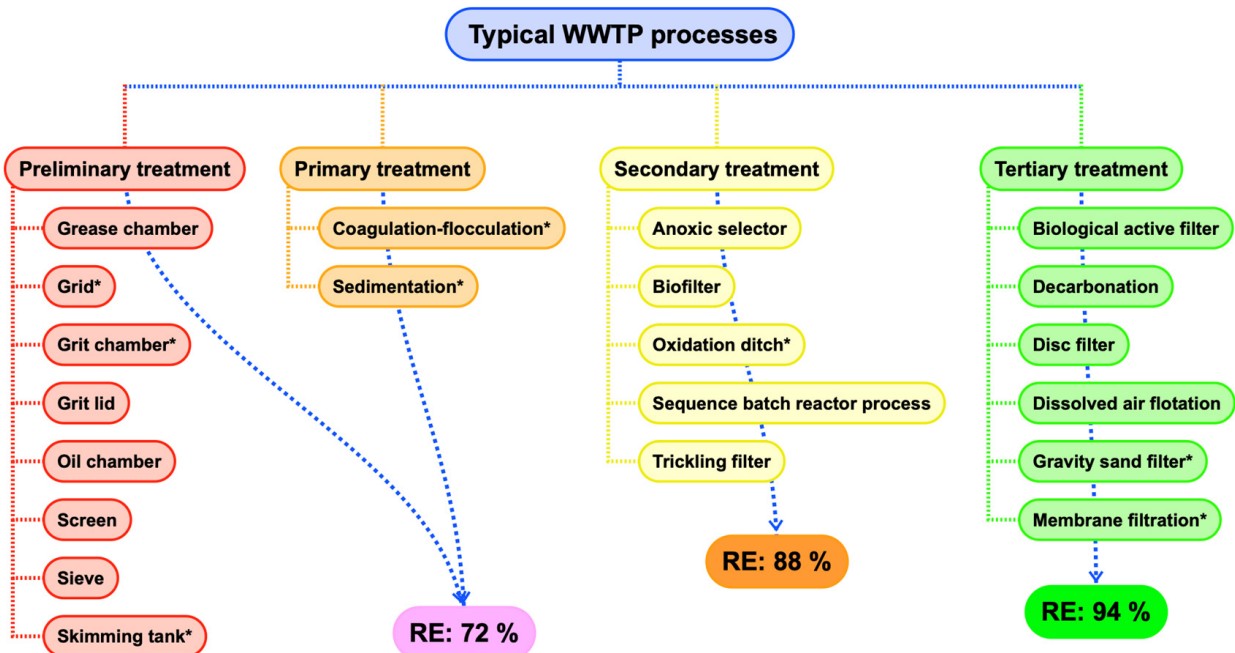

**Figure 1.** Main systems present in a wastewater-treatment plant (WWTP) [3]. Rough data on the removal efficiency (RE) of MPs of the different stages of treatment as a global, from Iyare et al. [4]. Those processes marked with an asterisk are present in the majority of plants.

If the treatment is focused on the purification of riverwater or water from desalination plants for human consumption, water-quality standards are much more stringent due to the expected lower concentration of foreign substances in drinking water. As a result, the preparation of drinking water generally requires the use of advanced water-treatment processes, such as membrane filtration and advanced oxidation. In fact, water and wastewater-treatment processes are similar, except for the biological processes that are required for wastewater due to the high content of organic matter [5].

Wastewater-treatment systems produce sludge, which must be treated before disposal or reuse [6,7]. The sludge needs to be concentrated to avoid the transport of extremely wet materials, among other reasons, and must also go through a process in order to destroy pathogens, and collect by-products such as biogas, in order to dispose of the sludge in an ecologically acceptable way [8].

On the other hand, as is well-known, plastics are present in almost all items related to human life, such as clothing, cosmetics, food packaging, equipment and instruments, etc. The annual global production of plastics has increased considerably in recent years, reaching 367 million tonnes in 2020 (only 1 million tonnes less than in 2019) [9]. It is important to add to these numbers the manufacture of synthetic fibers, since they are used in textiles, cords, fabrics, and other products, representing 61 million tonnes per year.

According to some estimates, around 10 million tonnes of plastic enter the oceans every year [10], accounting for 1.5 kg of plastic for each person on this planet, each year. Note that most plastics end up in sinks such as water collectors, rivers and seas until they drain into the oceans [11].

However, the problem of plastic is no longer that of "macroplastics" but of the plastic wastes being debased and reduced in size until they form what are called "microplastics", i.e., tiny particles. Plastics, although resistant, are exposed to various factors that cause them to break into smaller pieces, producing microparticles that are present globally.

Microplastics (MPs) are generally understood to be plastic waste with sizes smaller than 5 mm [10]. The effect of human exposure to MPs is not yet known, which raises many unresolved questions [12]. For this reason, several studies are currently being carried out on the presence and influence of MPs, especially in the marine food chain, since when MPs

reach the sea, they can be consumed by organisms such as zooplankton, thus entering the food chain.

Today, MPs are present everywhere, and come mainly from single-use plastics, fishing gear, clothing and cosmetics, paints, tires and urban dust [13]. These MPs easily reach rivers and seas due to their generally low density.

So far, three possible toxic effects of MPs have been indicated: the plastic particles themselves, their ability to adsorb persistent organic pollutants (POPs) and other substances present in aquatic systems around the world and the additives that these materials contain. Namely, the impact on human health stems from the fact that they can carry potentially toxic chemicals and microorganisms [14–17]. MPs could adsorb (or absorb) these dangerous chemicals, such as polyaromatic hydrocarbons (PAHs), metals and dioxins, from the environment and transport them to food products, which are ultimately consumed by humans [3,4,9,17].

MPs are classified as primary and secondary, as well as according to the polymer that constitutes them: polyethylene (PE), polyvinyl chloride (PVC), etc. Primary MPs are those that are already manufactured as MPs. More restrictions now apply to their manufacturing, but they have generally been used in hygiene products (exfoliating creams, toothpaste, gels, . . . usually known as personal care and cosmetic products, or PCCPs). Secondary MPs are those derived from the wear or degradation of large plastics (heat, UV, mechanical or biological degradation) [13,18]. Similarly, nanoplastics, with a size between 1 and 100 nm (i.e., 0.001–0.1 µm), can be produced by degradation of MPs or can be derived directly from industrial or domestic activity.

MPs are also classified according to their shape, and fall into the following categories (see Figure 2): fragments or microflakes (larger pieces, they could almost be called macroplastics); microfibers (with an elongated shape when observed under a microscope); microspheres or microbeads (which are small hard spheres); foams (sponge-like mass) and nurdles or granules (usually used in manufacturing). The terms nurdles and granules are used in many types of MPs when they do not fit in any of the others. The differences between pellets, nurdles, microbeads and microspheres are almost imperceptible. In much research, only the distinction between elongated microfibers and microparticles of a spherical or undefined shape is considered.

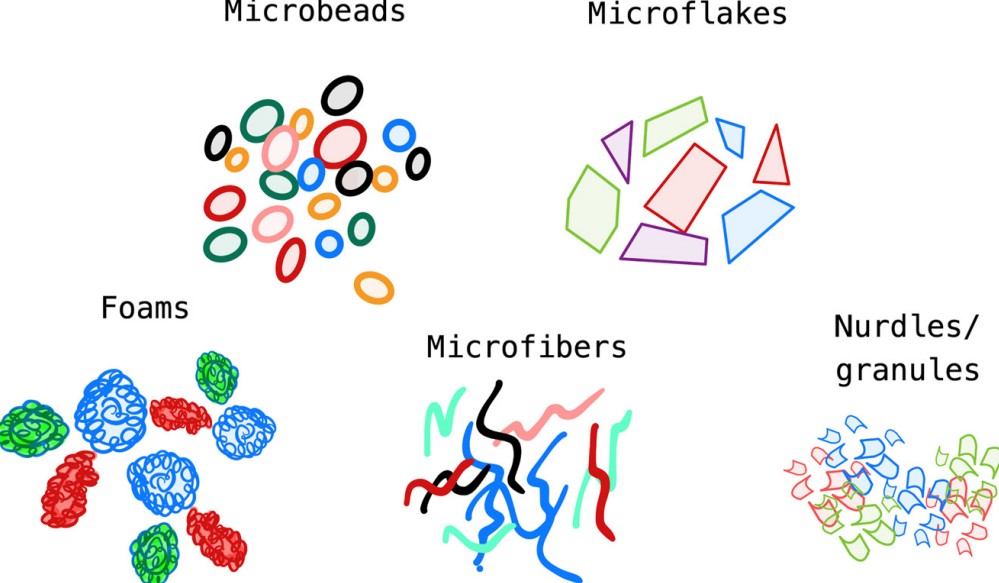

**Figure 2.** Main shapes presented by microplastics.

Each type of microplastic is associated with a main source. For example, microspheres are related mainly to the production of PCCPs, and microfibers are related to actions on synthetic polymers in clothing. These are considered the most common examples of

microplastic contamination in the environment [14]. Fragments and fibers are the two dominant shapes of plastic particles in wastewater [15].

Nowadays, one of the sources of MPs entering the ocean are treated wastewaters. According to Bayo et al. [19] the effluent of a treatment plant can vary between 5900 MPs/m$^{-3}$ on wet-weather days to 3000 MPs/m$^{-3}$ on dry ones. As commented above, many synthetic-fiber fabrics (e.g., nylon, polyesters, polyacrylamides, etc.) derive from washing processes [15]. In this way, textile fibers, which account for around 70% of the MPs in WWTPs, are also commonly found in wastewater. Many clothes are made of polyester, which is a plastic very similar to polyethylene terephthalate (PET) (that of the majority of water bottles, for example) and it is already in the form of fibers, demonstrating the huge potential problem they can pose [10,16].

Murphy et al. [20] pointed out the problem represented by WWTP, which can be considered as sources of MPs in the aquatic environment. The study, performed in a plant in Glasgow, showed that the removal efficiency of MPs was higher than 98%, but still the release of 65 million MPs into the receiving water was denounced.

In the present work, a review of the main findings related to the elimination of MPs in wastewaters is carried out, with special interest in elimination by filtration. The removal efficiency of several systems will be discussed, as well as the effect of the shape and size of the MPs to be eliminated and the nature of the polymers.

## 2. Microplastic Fate in Conventional Water/Wastewater-Treatment Processes

The fate of MPs entering WWTPs is currently a very relevant research topic. Many authors have investigated the role of MPs in the different stages of wastewater treatment. It has been observed that the presence of MPs reduces the efficiency of wastewater and sludge-treatment processes and increases sludge volume [6,21], affecting methane production in bioreactors, for instance. The presence of MPs could make it necessary to change the configuration of the treatment plant. For example, the presence of a large number of MPs could oblige to the use of certain reagents. Similarly, a decrease in the nitrogen conversion rate and fouling of the membrane in wastewater-treatment processes have been described [22]. As a result, it is difficult to accurately assess MP-removal efficiency (RE) in WWTPs. At present, there are no policies or legislation that require a minimum for MP removal during wastewater treatment.

Johnson et al. [23] looked for the presence of MPs in potable water. They found extremely low amounts, less than 2 particles/m$^3$, showing that there was an effective removal of this type of particles in the water-treatment plants.

Table 1 shows a summary of the most important data summarized in the present review.

In their review, Liu et al. [24] presented the numbers of MPs in the influent of different WWTPs. They showed that the abundance of MPs in the influent was between 0.28 and 31,400 MPs/L, while that in the effluent was between 0.01 and 297 MPs/L. They found quantities of MPs in the sludge varying within the range $4.40 \cdot 10^3$–$2.40 \cdot 10^5$ MPs/kg.

Rasmussen et al. [25] calculated the masses of plastics of >500 μm in a WWTP. They showed that the plant load was 202.2 kg/day, of which 73% was retained. The remaining plastic was found in the sludge (13.6%) and in the effluent (0.4%).

Tadsuwan et al. [26] pointed out that the RE of a grit trap alone was >33%, followed by 14.2% removal in a secondary treatment. In the sludge, the number of MPs was 8120 particles/kg.

Even considering that conventional WWTPs usually eliminate more than 90% of the incoming MPs, these systems continue to be the main source of introduction of MPs into the environment, as the high volumes of generated effluents represent a very large contribution [27]. Simon et al. [28] noted that Danish WWTPs discharge around 3 t/year of MPs in the size range of 10–500 μm, despite the 98% RE of this size of particles. Bretas-Alvim et al. [27] presented data on the discharge of MPs in different countries, observing important differences ($\pm 1000\%$ approx.) among them.

Recently [29] a study investigating the MP removal at different treatment stages showed that the primary treatment process has an effectiveness of about 75%, and that it increases to 91.9% in the secondary stage. Removal efficiency further increased to >98% after tertiary treatment (coagulation, ozonation, disc membrane filters and rapid sand filters).

Iyare et al. [4] studied the fate of MPs in 21 WWTPs around the world. The authors show that the preliminary and primary treatment of wastewater remove 72% of the MPs, and that the removal efficiency increases to 88% (of the number of microplastic particles present in sewage influent) in secondary treatment and 94% in the tertiary. A major part of the MPs eliminated are accumulated in the sludge, which has lately been used for soil enrichment, causing accumulation in terrestrial ecosystems [4]. Ou and Zeng [30] pointed out the minor presence of MPs at night, showing an uneven temporal distribution of MPs.

The largest WWTP in P. R. China was studied by Yang et al. [31]. A concentration of approx. 12 MPs/L was detected in the influent, reducing the amount to 0.6 MPs/L by employing a conventional WWTP that was equipped with ultrafiltration as the final tertiary treatment. This study shows a decrease in the MP concentration throughout the treatment processes, i.e., the RE in the primary treatment was ca. 59%; it was 72% after the secondary stage; and that of the overall process was 95%. This research also investigates the role of the MP size and found an entering average size of ca. 1110 μm, being quite similar to the size in the effluent. The main polymers detected, as in many other studies, were PET, PS (polystyrene) and PP (polypropylene). The fact that they find similar sizes at the inlet and the outlet of the plant after an ultrafiltration process suggests that MPs might not be properly retained in the filter, or that a mix of filtered and unfiltered waters was produced.

Nevertheless, other authors [32] observed a lower removal efficiency in WWTPs located in Thailand. In this work, the overall RE was between 67 and 57% in the dry and rainy sessions, respectively.

Recently, Wu et al. [22] pointed out that secondary treatment was, within the WWTP, the most efficient process to eliminate MPs from wastewater. According to their data, the removal efficiency can be as high as 98%.

**Table 1.** Summary of data found in literature. MBR: membrane bioreactor; RSF: rapid sand filter; UF: ultrafiltration; WWTP: wastewater-treatment plant; DWTP: drinking-water-treatment plant.

| Country | Influent | Treatment | Effluent | Reference |
|---|---|---|---|---|
| Spain | 4.40 MPs/L<br>4.40 MPs/L | MBR<br>RSF | 0.92 MPs/L, average size = 1.39 mm<br>1.08 MPs/L, average size = 1.15 mm | Bayo et al. [18] |
| Spain | | Global | Effluent with 5.9 MPs/L in wet weather and 3.0 MPs/L in dry ones | Bayo et al. [19] |
| Israel | 74% fibrous MPs | Global<br>RSF<br>RSF | RE = 97%<br>1.97 MPs/L<br>91% fibrous MPs | Ben-David et al. [33] |
| Spain | 90% fibers (9% MPs) | | | Bretas-Alvim et al. [27] |
| USA | | Primary and secondary | 0.00144 MPs/L | Carr et al. [34] |
| (several) | 61–5600 μg/L | Global<br>Global | RE > 93%, by weight<br>0.5–170 μg/L | Cheng et al. [3] |
| Italy | | Global | RE > 90% | Cristaldi et al. [15] |
| Spain | | RSF | RE > 78% in DWTP | Dalmau-Soler et al. [35] |
| France | | Primary<br>Primary and secondary | RE = 80% for fibrous materials<br>RE = 95% for fibrous materials | Dris et al. [36] |
| | | Global | Up to 88% of MPs removed go to sludge | Freeman et al. [37] |
| Israel | 32.4% suspected MPs were plastics | Primary and secondary | 65.6% fibers, 28.1% fragments, 5.4% pellets | Gies et al. [17] |
| Greece | 3160 MPs/L | Global<br>Global<br>RSF | RE = 97–99%<br>125 MPs/L<br>RE = 98.9% | Gatidou et al. [38] |
| South Korea | | Primary<br>Secondary | RE = 79%<br>RE = 92% | Hidayaturrahman et al. [29] |
| (several) | | Preliminary and primary<br>Secondary<br>Tertiary | RE = 72%<br>RE = 88%<br>RE = 94% | Iyare et al. [4] |
| China | 126 MPs/L | Global | 30,6 MPs/L in the effluent<br>>75% of MPs transferred to the sludge<br>Accumulation of 7.74·10$^{12}$ particles/year in the sludges. | Jiang et al. [39] |

**Table 1.** *Cont.*

| Country | Influent | Treatment | Effluent | Reference |
|---|---|---|---|---|
| Thailand | | Global | RE = 67% (drain season) | Kankanige [32] |
| | | Global | RE = 57% (dry season) | |
| | | Global | RE = 60% for sizes 6.5–53 μm | |
| | | Global | RE = 80% for size > 500 μm | |
| | | RSF | RE = 26.8% for the smallest particles | |
| | | RSF | RE = 60.7% for the biggest particles | |
| Singapore | | Membrane filter | RE < 7% for nanoparticles (100–500 nm) | Li et al. [40] |
| Denmark | 0.917 MPs/L | Biofilter | 0.179 MPs/L | Liu et al. [41] |
| | | | RE = 100% for sizes lower than 0.1 mm | |
| Denmark | 0.28–31,400 MPs/L | Global | 0.01–297 MPs/L | Liu et al. [24] |
| China | | Primary | RE = 75% | Lv et al. [42] |
| | | Secondary | RE = 95% | |
| | | Tertiary | RE = 98% | |
| | | MBR | RE = 99.5% | |
| | | Activated sludge | RE = 97% | |
| USA | | Global | Effluent with less than 1 particle per liter | Mason et al. [43] |
| | | | $5 \cdot 10^4$–$1.5 \cdot 10^7$ particles/day | |
| USA | | Global | 60% fibers in the effluent | Michielssen et al. [44] |
| Germany | | Use of magnetic ionic liquids | RE = 100% (PS) | Misra et al. [45] |
| Canada | | Filtration with 0.22 μm | RE > 99% | Murray et al. [46] |
| Scotland | 15.7 MPs/L | Global | 0.25 MPs/L (RE > 98%) | Murphy et al. [20] |
| China | | (Simulated runs) | Smaller-sized MPs are more mobile in RSF | O'Connor et al. [47] |
| | | | A greater number of wet–dry cycles increases MPs' penetration depth | |
| China | | | Grease on the sludge contains many more MPs than grit and sludge cake. | Ou and Zeng [30] |
| Sweden | 202.2 kg MPs/day | Global | RE = 72% | Rasmussen et al. [25] |
| Germany | | Seeds of $Fe_3O_4$ and application of magnetic field | 95% | Rhein et al. [48] |
| Indonesia | | RSF | RE = 50% for plastic flakes, in DWTP | Sembiring et al. [49] |
| | | | RE = 96% for tires, in DWTP | |
| India | | RSF | RE = 85%, by weight | Seth et al. [50] |
| | | | RE = 90%, by number | |
| Denmark | | Disk filter | RE = 76% | Simon et al. [51] |
| Denmark | | | 3 tonnes MPs/year, size range 10–500 um, 98% RE | Simon et al. [28] |
| Thailand | | Grit trap | RE > 33% | Tadsuwan et al. [26] |
| | | Secondary treatment | RE > 14.2% (additional) | |
| (several) | | Primary | RE = 98% | Talvitie et al. [52] |
| | | Primary and secondary | RE > 99% | |
| | | Secondary | RE = 88% | |
| | | MBR | RE = 99.9% | |
| | | RSF | RE = 97% | |
| | | Dissolved air flotation | RE = 95% | |
| | | Disc filtration | RE = 40–98% | |
| Spain | | | SS has a lighter plastic load, much lower than heavy plastic load (18,000 vs 32,070 MPs/kg) | Van der Berg et al. [53] |
| China | 12 MPs/L | Primary and secondary | RE > 98% | Wu et al. [22] |
| | | UF | 0.6 MPs/L | |
| China | | Primary | RE = 59% | Yang et al. [31] |
| | | Primary and secondary | RE = 72% | |
| | | Global | RE = 95% | |
| India | | Hollow-fiber membrane | RE = 99.3% | Yaranal et al. [54] |
| USA | | RSF | RE = 86.9% for small particles | Zhang et al. [6] |
| | | RSF | RE = 87% for critical size (10–20 μm) | |
| | | RSF | RE > 99.9% for particles larger than 100 μm | |

Talvitie et al. [52] investigated the behavior of advanced treatment technologies in various WWTPs. The highest removal efficiency was achieved by the membrane bioreactor (MBR) with a RE of 99.9%, followed by gravity or rapid sand filter (RSF) (RE = 97%) and air flotation (RE = 95%).

In the study led by Lv et al. [42] two parallel systems of wastewater treatment were studied, namely oxidation ditch treatment and MBR. The MPs observed by these authors at the plant entrance consisted of PET (47%), PS (20%), PE (18%) and PP (15%). Regarding the shape of the microparticles, a greater proportion of fragments was found (65%), followed by fibers (21%, mainly PET), films (12%) and foams (2%). No microbeads were observed. Regarding the size of the particles, the dominant size was >500 μm (40%) and between 62.5–125 μm (29%). These researchers observed that MP concentrations increased in the

treatment systems as a function of the ease of the treatment process. In this way, the MPs were eliminated by 99.5% in the MBR system compared to 97% in the oxidation system (by weight). These percentages indicate the tendency to retain the largest fragments.

The study developed in Finland by Talvitie et al. [55] also showed the effectiveness of WWTPs for the elimination of MPs. The results show that conventional primary and secondary treatments can effectively eliminate (>99%) the MPs that reach the WWTP. The majority (98%) of the MPs were eliminated during the primary treatment. The activated sludge (secondary) process further decreased (~88% from process input) the MP concentration. During wastewater treatment, most MPs (>99.5%) were retained in the primary and secondary sludge; however, part (~20%) of the retained MPs was recirculated back to the treatment process as part of the water was rejected.

The authors [55] stated that the removal of MPs can be further improved with the implementation of a specific treatment system. The proposed equipment consists of a membrane bioreactor (additional RE = 99.9%), sand filtration (RE = 97%), dissolved air flotation (RE = 95%), or disc filtration (RE = 40–98.5%). Biologically active filtration had no impact on the concentration of MPs. Nevertheless, Bayo et al. [12] found persistent low-density polyethylene (LDPE), nylon and PVC in the RSF and melamine after MBR treatment.

Gies et al. [17] studied the presence and removal of MPs in a Canadian WWTP. They found that only 32.4% of the suspected MPs were plastic polymers, and that the global removal efficiency of the plant varied between 97.1% and 99.1%. Nevertheless, the number of MPs released to the environment was concentrated in the primary and secondary sludges.

Ben-David et al. [33] studied the elimination of MPs in a WWTP sited in northern Israel, observing a removal efficiency higher than 97%, with most MPs eliminated in the secondary stage and observing an average of 1.97 MPs/L after sand filtration.

Carr et al. [34] showed that existing treatment processes are effective at removing MPs in WWTPs, and found no MP presence after tertiary treatment in seven plants, and only 0.00114 MPs/L for effluent after secondary treatment. In this way, MP particles were found to be removed mainly in the primary treatment zones during the processes of solid skimming and sludge settling. These authors mention that some consumer products, as the toothpaste containing MPs may be contributing much more than others to the amounts of MPs reaching the WWTPs.

Cheng et al. [3] found a global removal efficiency of MPs higher than 93%, by weight, going from 61–5600 µg/L to 0.5–170 µg/L. These researchers showed that preliminary and primary treatments contribute to the removal of MPs, and that the efficacy of secondary and tertiary treatments was, understandably, highly dependent on the applied techniques.

Cristaldi et al. [15] reviewed information on the removal efficiency of different WWTPs around the world, showing a removal efficiency greater than 90%. This seems to be a good situation, but the high amount of water treated in the plants means that millions of MPs continue to be released every day into the aquatic environment.

Biofilters have also been tested for the removal of MPs from treated wastewater. In their study, Liu et al. [41] showed that a raw effluent containing 0.917 MPs/L was easily filtered to 0.179 MPs/L in the first stage of a conventional biofilter. Nevertheless, small sizes, as expected, cannot be completely removed, although no MPs higher than 100 µm in size were found (i.e., RE = 100% for sizes lower than 0.1 mm).

Ngo et al. [56] showed that the existing WWTPs are inefficient to completely remove the MPs and there is a risk that they may be discharged into the ambient water.

In many studies, filtration has been shown to play an important role in removing MPs. However, this method has its drawbacks [21,57] because the mechanical stress generated during treatment can cause plastics to wear away, resulting in smaller particles that are likely to be released into the environment without restriction.

## 2.1. Microplastic Accumulation in Sludges

Zhang et al. [6] insisted that in a WWTP, microplastics are simply transferred from sewage to sludge; this could be a problem during anaerobic digestion, as MPs can contain

significant amounts of toxic substances such as persistent organic pollutants. Gatidou et al. [38] showed that MP content ranges up to 3160 MPs/L in raw water, 125 MPs/L in treated wastewater and $170.9 \cdot 10^3$ MPs/kg total dry solid in the sludge. Gies et al. [17] observed that primary sludge had approx. $14.9 \cdot 10^3$ MPs/kg, with fibers being more abundant than other particles. Likewise, secondary sludge had $4.4 \cdot 10^3$ MPs/kg, with fibers also being the most abundant shape. The dominance of fibers in sludge samples has been previously reported [58,59]. On their hand, Liu et al. [24] indicated that the microplastic abundance in the sludge was within the range of $4.40 \cdot 10^3 - 2.40 \cdot 10^5$ particles/kg.

Because MPs accumulate in biological sludge, and because these sludges are mainly used to improve agricultural or forest soil, studies should be carried out on the incorporation of these particles in meat from land animals [27].

Regarding the MP content in sewage sludges (SS), Ou and Zeng [30] indicated that the grease on the SS samples contained a number of MPs that was significantly higher than the grit and sludge-cake samples. PE microbeads from PCCPs were dominant in grease samples.

Habib et al. [60] observed an abundance of synthetic fibers in soil conditioners, fertilizers and similar biosolid materials, denouncing the presence of MPs in sludges.

Jiang et al. [39] studied the fate of MPs in sewage and sludge filter cake. These authors showed that there were 126.0 MP particles/L in the influent and 30.6 in the effluent, transferring about 75% of the MPs to the sludge. In their study, the abundance of MPs in dewatered sludge and filter cake was 36.3 and 46.3 particles/g. These figures are equivalent to the accumulation of $7.74 \cdot 10^{12}$ particles/year in the sludges.

This implies that most of the MPs removed during wastewater treatment will end up in the open environment [33,61]. For this reason, Wolff et al. [62] indicated that WWTPs are a good sink for MPs only when the sludge is thermally treated and not used for agricultural amendment. Incineration and production of chemicals by pyrolysis of sludges from WWTPs are very promising current lines of research [63] that would avoid the accumulation of MPs in the environment. These techniques would, furthermore, reduce the energetic demand on the treatment plants, increasing the potential for chemical-energy recovery from wastewater pollutants [64].

In this way, an alternative route for disposal, incineration of SS with heat recovery, has been pointed out as a good practice [65]. Note that in the European legislation [66] incineration is preferred to landfilling. This is because landfill is much less controllable, producing gas release and in many cases spontaneous combustion of the biogas, in addition to the fact that the necessary technology for the treatment of the gas produced in the incinerator is already available. Through incineration, MPs contained in the SS would definitely be eliminated from the environment, in what emerges as a safe alternative for disposal.

### 2.2. Need for Standardized Methods

There is no standard protocol for MP measurement in WWTPs. Many authors [3,27,67–70] insist that the differences between studies are related to the different analytical techniques employed for measuring MP contamination.

Lusher et al. [71] insisted on the necessity of harmonization in the isolation and extraction of MPs from environmental samples, including the sample-collection techniques and data analysis. On the other hand, Oβmann [72] insisted that the harmonization of methods for the quantification of MPs in drinking water and other food should be developed and defined.

Bayo et al. [19] indicated that the concentrations reported by different authors on the presence of MPs in WWTPs are difficult to compare, because different sampling techniques, volumes and analytical methods are used. Furthermore, the use of MP concentrations on a numerical basis has been criticized for overstating MP abundance because this involves counting the number of suspected particles [3].

Sun [73] pointed out that since MP sampling and detection methods can significantly affect the result of their quantification and identification, harmonization is urgent. Hong et al. [74] proposed a method for estimating the mass of MPs in sewage, based on the quantification of total organic carbon (TOC).

Moreover, differences can be found due to the difficulties present in the sampling of MPs in waters. Lenz et al. [75] designed a water-sampling system specially dedicated to the collection of MPs in surface and groundwater for environmental pollution studies. It is designed specifically for sampling small MPs (size fraction <300 μm) in different source areas. Li et al. [40] mentioned that in many MP studies, these particles are just sampled by nets with a typical mesh size of 330 μm. The sampled MPs are usually purified and digested, and lately analyzed by different methods, but there are no universally accepted sampling and quantification tools, so important differences among studies can be found.

Another interesting topic is the treatment of samples in the laboratory. Elkhatib et al. [69] insisted on the need for quality-assurance methods. This would include the use of a wet filter exposed to air during sample processing to take into account air contamination, the analysis of blank samples using deionized water, the addition of a sample with a known polymer, and the analysis of samples in triplicate. Ou and Zeng [30] showed that the use of $H_2O_2$ to preoxidize the particles can remove surface biogenic materials from the polymer. However, the preoxidation combined with heating induced the melting of some MPs, especially microbeads. Dyachenko et al. [76] also used peroxide oxidation to eliminate interferences in the analysis.

Gatidou et al. [38] also insisted that special care should be taken on the types of clothing used. Gies et al. [17] observed a deposition of 36 fibers on a 1 μm pore-size × 47 mm-diameter filter membrane over an 8 h period.

## 3. Filtration as a Route of Elimination

Liu et al. [24] compared the performance of 38 WWTPs in different countries and showed that filter-based treatment processes attained the highest microplastic-removal efficiency.

In the RSF, the sand bed captures the particles when the secondary wastewater effluent passes on, and the MPs can be removed by backwashing the sand; this process can remove as much as 97% of the MPs from the wastewater [37]. Bayo et al. [12] discussed the role of two different treatment technologies in the abatement of MPs, showing a RE of ca. 80% for MBR and of 76% for RSF.

The study by Hidayaturrahman and Lee [29] used a sand bed for treating wastewater from the coagulation process in one sand filter by gravity. In this study, the MP retention rate was lower compared to ozone and membrane disc-filter technologies, but the efficiency was still 98.9%. It was observed that MPs whose dimensions are smaller than the diameter of the sand are more likely to pass through the filter.

In the work published by Seth and Shriwastav [50], reference was also made to the filtration of waters with a high content of MPs. In this case, a sand filter was used with water that had been artificially contaminated, with removal efficiencies of around 85% by weight, which represents 90% by number. The authors tested nine different commercial polymers. MPs were quantified before and after sand to establish filtration efficiency. For all thicknesses of the sand bed, the removal efficiency (% by weight) was greater than 85% and continued to improve with subsequent cycles, while it remained at 87% after 25 cycles. The RE of particles of different sizes was also estimated, finding RE (% by number) higher than 90% for all sizes, although larger particles (>500 μm) showed a higher filtration efficiency.

In the study by O'Connor et al. [47] the vertical mobility of MPs in sand columns was investigated. The study focused on the penetration of rain-drying cycles in a terrain exposed to climatic variations, but it may be interesting from the point of view of the efficiency of filtration in a sand bed. The study carried out experiments in a column of sandy soil, checking the mobility of five different MPs, which consisted of PE and PP particles of various sizes and densities. The experiments were mounted on a transparent PVC column and several various amounts of liquid were fed to the column, with the

objective of examining the effect of the filtration volume on the migration of MPs. The study revealed that the smallest PE were those with the greatest mobility among the five MPs evaluated. It was also concluded that subjecting these MPs to a greater number of wet–dry cycles increased their depth of penetration in the sand columns, with a practically linear relationship between the depth of penetration and the number of the wet–dry cycles; in comparison, almost no variations in the migration of the microparticles were observed when the volume of the filtration liquid or the initial concentration of MPs in the column varied. In this study, the quantities of MPs introduced into the bed were very high compared to the other studies mentioned, which may bias the results.

Michielssen et al. [44] showed that efficiency of a WWTP was high (>95%) in removing small anthropogenic litter, but this litter consisted of many different materials, including cellulosic products manufactured from natural materials, among others. Indeed, the authors insist that tertiary granular-sand filtration exhibited a high RE for the different materials and recommend the installation of this device in the WWTPs, mentioning a possible decrease of more than 50% in the presence of MPs. On the contrary, Carr et al. [34] found that WWTP using RSF as a tertiary treatment showed low removal efficiency of MPs.

Other authors maintained that MPs can be adsorbed by silica grains due to hydrophilic interaction, thereby clogging the initial layers and reducing RSF performance faster than expected. Thus, even though backwashing is performed regularly, MP adsorption can be difficult to reverse due to the presence of hydroxyl groups on the surface of MPs as a result of weathering [34,77]. Different new techniques with high potential for removing MPs have been tested, such as dynamic membranes, enhanced flocculation/coagulation systems and electrocoagulation.

RSF has also been used to remove MPs from drinking water. Sarkar et al. [78] tested a drinking-water facility for the removal of MPs, indicating that the influent has a very little amount of MPs (17.8 MPs/L), and that pulse clarification and sand filtration removed 63 and 85% of the particles, respectively. Dalmau-Soler et al. [35] studied the removal of MPs in a DWTP close to Barcelona (Spain). In this research, the authors showed a predominance of PS and PP in the intake water from a river, with a mean concentration of 0.96 MPs/L, while at the outlet it was $0.06 \pm 0.04$ MPs/L, with an overall removal efficiency of 93%. The authors mention the importance of the sand filtration stage, calculating a mean removal efficiency of 78%. This research points out the effectivity of methods such as ultrafiltration/reverse osmosis, much more effective than ozonation/carbon filtration stages in the MP removal.

Sembiring et al. [49] performed tests on the RSF of drinking water containing MPs of plastic from shopping bags (probably PE) and tires that were crushed using a grinder. The produced particles were 10 to 500 μm in size and the authors found a RE of about 50% for plastic flakes and about 96% for tire particles. For the study, filtration rates between 4 and 10 m/h were used, and the runs were performed in a RSF unit constructed ad hoc. Samples were taken at different times in order to study the possible differences produced by fouling of the filtration media. The authors also studied the effect of the filter media porosity on the RE for both kinds of MPs, comparing the RE obtained for the effective size of the filtration media of 0.39 mm and 0.68 mm. The authors understandably found a decrease in the RE as the size of the filtration media increased, but in all runs the particles bigger than 200 μm were completely removed.

Other filtration techniques have also been used for MP removal. A disk filter was studied by Simon et al. [51], where they captured ca. 76% of the particles contained in the secondary effluent of a WWTP.

Ding et al. [79] showed the great effectivity of membrane filters for the removal of MPs, but discussed the possibility of releasing nano- and microplastics from organic membranes in drinking-water-treatment plants. They insist on this possibility and demand for more research on this topic.

Ahmed et al. [80] reported that membrane technologies such as UF, microfiltration and MBR increase the removal efficiency of MPs and nanoparticles.

On the other hand, Li et al. [81] showed that many nanoparticles are not retained by membrane filters with a pore size as small as 1 µm. In their runs, using spherical PS and poly(methyl methacrylate) (PMMA) plastics as models, the efficiency of membrane filtration was lower than 7% for nanoparticles (size of 100–500 nm). These authors propose cloud-point extraction to concentrate these small particles for analysis.

Finally, an interesting study was published by Rhein et al. [48]. The authors proposed a magnetic seeded filtration to improve the RE of MPs. The study uses PVC and PMMA particles of a very low diameter (mean diameters of 2.06 and 5.98 µm, respectively), together with seeds of $Fe_3O_4$. The application of a magnetic field seems to efficiently improve the removal of these very small particles (RE > 95%), which are usually problematic. Rhein et al. determined the optimal working conditions studying pH, intensity, rotary speed and concentration of particles. Misra et al. [45] also studied the removal of MPs using magnetic techniques, in the case of ionic liquid phases, that furthermore present bactericidal properties. The removal efficiency of their system was 100% for tested PS beads.

## 4. Differences among MPs Being Filtrated

The size and shape of the MPs being eliminated can play an important role, as well as the nature of the polymer that makes up the plastic. It has been indicated in literature that the morphological characteristic is one of the most important factors that contribute to the removal process. This is mainly due to the interaction between plastic waste and other pollutants, as well as microorganisms in wastewater [56]. Murray et al. [46] studied the RE for nanoplastics (<400 nm) in water and sewage-treatment plants, showing that filtration with a 0.22 µm filter removed 99% of the particles, and ballasted flocculation removed 88%.

Fibers and fragments are the two dominant types of plastic particles in wastewater [15]. In many cases, MP fragments are associated to the presence of PE and PP (63% and 25%, respectively [27]), whereas microfibers related to PE probably originated in washing machines. The type of polymer most present was PE in almost all WWTPs investigated [15].

Gies et al. [17] indicated that in the primary and secondary effluent-wastewater samples, the presence of MPs was dominated by fibers (65.6%), followed by fragments (28.1%) and pellets (5.4%), while only a few were classified as foams, granules and sheets. (0.22%, 0.45% and 0.20%, respectively).

Bayo et al. [82] calculated a RE for MPs of 64.22% after the tertiary treatment, one of the lowest figures found in the literature. Fibers were less-efficiently removed (56.16%) than particulate MPs (90.03%)

Different researchers point out that wastewater-treatment plants are capable of removing large amounts of larger MP particles but are ineffective at removing particles with any dimension of less than 100 µm, and tributaries and effluents tend to have similar amounts of these smaller particles [37]. In this sense, Michielssen et al. [44] calculated that the microparticles in the final effluents of two WWTPs in the USA included more than 60% fibers, with estimated concentrations of 3.58 and 5.25 microfibers/L and 1.94 and 0.8 fragments/L. Contrarily, Sun et al. [73] considered that the pretreatment in a WWTP was more effectively removing fibers than other types of MPs, and they found that the relative abundance of fibers decreased after the pretreatment.

Freeman et al. [37] affirmed that there are no methods capable of efficiently removing very small plastic particles and fibers in a way that is technically, environmentally and economically sustainable in wastewater-treatment plants.

Bretas-Alvim et al. [27] investigated the presence of different types of microparticles in a WWTP sited in Valencia (Spain), finding a much higher presence of microfibers than other shapes (90%), but only 9% of these were confirmed to be MPs.

Hu et al. [14] showed that most MPs from PCCPs (usually in the shape of microballs) can be efficiently removed during wastewater treatment, although the efficiency was not so high for MPs in the shape of fibers, mainly coming from the domestic washing of synthetic fibers.

Furthermore, Talvitie et al. [52] reported that the primary sedimentation acts mainly on textile fibers, while the contribution of this step to the elimination of synthetic particles was almost negligible. Similarly, Liu et al. [24] mentioned that primary treatment has superiority over secondary and tertiary treatments for plastic-fiber removal, indicating that fibers were easily trapped during primary treatment due to flocculation and settling. On the other hand, fragments exhibited excellent removal efficiency during the secondary treatment process. Dris et al. [83] observed a RE > 85% for MPs in the form of fibers during primary treatment, while this efficiency increased to almost 95% after biological treatment.

Dalmau-Soler et al. [35] linked the MPs in the shape of fibers to PS and PP, and the shape of fragments to particles of PE. The presence of PS fibers was related to the washing of synthetic clothes. The authors found a slight decrease in fibers of 20–500 μm after sand filtration but with an increase in larger fibers thereafter.

Simon et al. [28] centered their study in the shape and weight of the MPs found in a WWTP. They found no significant differences between the mass of the particles in the wastewater samples before and after the treatment, showing a median particle mass of 13.45 ng in raw wastewater and 3.52 ng in treated wastewater. Furthermore, they insisted that research should focus on both the length and the diameter of the fibers, and mentioned that a single dimension is not representative of the features of the particles. For fibers with small diameters (<50 μm), a single dimension could not adequately define size or provide sufficient information on its contribution to the total number of MPs in a sample.

In a very recent study, Wolff et al. [62] showed that the presence of an RSF removed on average around 99.2% of the total discharge of MPs, in experiments where the majority of the particles had a diameter of <100 μm. The authors showed that after sand filtration, PE was the only polymer found among the MPs. All particles >500 μm were removed by the sand filter and the predominant one in the RSF effluent was 10–50 μm in size.

Ben-David et al. [33] observed a lower efficiency for the removal of fibers in a sand filtration unit, representing fibers in 74% of MPs in the influent, and increasing to 91% after filtration.

Kankanike et al. [32] found a majority of fibers in the treated waters, with PE, PP, PET and PVC commonly identified. These authors showed removal efficiency was a function of the MP size, as expected. RE increased with MP size, going from 60.0% (dry season) for a size between 6.5–53 μm (46.9% in the rainy season), to 80.2% (dry season) for a size >500 μm (67.5% rainy season). With respect to the filtration step, Kankanike also showed an increase in RE with size, but very low values were found in comparison to other studies. In this case, reported RE for filtration was 26.8% for the smallest particles (12.7% in the rainy season), and 60.7% (52.1% rainy) for the biggest. The filtration unit of the plant included two layers: one of anthracite coal as the top layer (500–600 μm porosity) and the bottom layer being graded sand (400–500 μm porosity). Both layers can retain ≥500 μm MPs, thus showing a good retaining efficiency for higher sizes. In contrast to filtration, Kankanike et al. showed that clarification presented a higher removability for smaller-sized particles [32].

With regard to the removal efficiency of granular filters, Zhang et al. [84] found that it was much more effective to filter micro- and nanoplastics (RE = 86.9% for small particles, RE >99.9% for particles larger than 100 μm); conversely, a critical size exists (10–20 μm) where RE was quite poor, namely 87%. This means that the RE is not proportional to MP size, contrarily to that found by many researchers, and a smaller size does not necessarily mean a lower RE.

Finally, Ma et al. [85] observed that coagulation processes are more effective in the elimination of smaller particles of PE compared to bigger ones. In addition, they obtained a complete removal of all PE particles by ultrafiltration, observing a slight membrane fouling due to the large diameter of MPs compared to the ultrafiltration-membrane pores.

## 5. Conclusions

A bibliographic review has been carried out on aspects related to the presence of MPs in water and wastewater-treatment plants, with special attention given to the efficiency of the filtration process in removing MPs.

Although in most current systems the removal of MPs is effective, important differences have been observed in the removal efficiencies of smaller plastics, which are not removed effectively. In addition, it has been shown that the filtration process can break up the particles, creating larger amounts of very small MPs.

The quantities of MPs eliminated from the waters mostly end in the sewage sludge. These are generally used for agricultural applications, which results in their dispersion through the environment in a very worrying way. Only thermal treatment of this sludge in well-designed systems will be a desirable process from the point of view of their disposal.

Even with large removal rates, the small quantities of MPs that remain in the effluents can pose a serious problem, given the large volumes of effluents that are discharged into the aquatic environment.

Some of the most important points can be summarized as follows:

— Although the removal efficiency of MPs in WWTP is relatively high, the quantity of these particles discharged into the environment is still very high, becoming the largest source of the introduction of MPs into the environment.
— Primary and secondary treatment processes effectively remove MPs from wastewater with removal efficiencies ranging from 75% to 91.9%. RE can be increased to >98% after tertiary treatment.
— Most of the MPs eliminated from sewage accumulate in the sewage sludge. Incineration/thermal treatment of SS is the only effective way to destroy these plastic particles.
— During sewage treatment, part of the retained MPs can be recirculated back to the treatment process as part of the water is rejected.
— There is a need for harmonization of the techniques used for MP analysis and counting.
— Larger particles exhibit greater filtration efficiency. Smaller particles have more mobility.
— RSF shows a good performance for MPs in the form of beads or granules. However, it does not guarantee microbeads (nor microfibers) would be absent from the effluent.
— Efficiency of membrane filtration can be as low as 7% for nanoparticles.

**Author Contributions:** Conceptualization, N.O. and J.A.C.; writing—original draft preparation, J.A.C.; writing—review and editing, N.O.; funding acquisition, J.A.C. All authors have read and agreed to the published version of the manuscript.

**Funding:** This research was funded by MINISTRY OF SCIENCE AND INNOVATION (Spain), grant number PID2019-105359RB-I00, and by the UNIVERSITY OF ALICANTE, grant number UAUSTI20-05.

**Institutional Review Board Statement:** Not applicable.

**Informed Consent Statement:** Not applicable.

**Data Availability Statement:** Not applicable.

**Conflicts of Interest:** The authors declare no conflict of interest.

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
