# Peer review of "Reuse of Water Contaminated by Microplastics, the Effectiveness of Filtration Processes: A Review"

_energies, doi:10.3390/en15072432_

Round 1

Reviewer 1 Report

This work is interesting. However, several modifications are needed prior to acceptance.

  1. This manuscript should be submitted to the proofreader.
  2. Table 1 is too general and does not contain a significant summary. Efficiency, method, advantages, limitation of the methods, any gaps in the cited work should be included in the table to give readers better clarity.
  3. Section 3 is like summarizing every single reference without having any thought or comments from authors on the cited works. The whole section is like reading an abstract of each paper only. Please improve this part.
  4. Please also put a section on the environmental impacts of these microplastic issues from the perspective of the Life Cycle Assessment study.
  5. Please relate these works with the current UN SDG target and what is the current progress of the researchers cited in this work and map it with the current SDG target. 

Reviewer 2 Report

General Comments

The authors conducted a long, extensive read and obtained valuable data. I appreciate the efforts of the authors. However, the current version of manuscript has some problems that cannot be ignored. To meet the standards of Journal, some revisions are necessary for this manuscript.

Specific Comments

In this article, tenses are not uniform. The present tense is mixed with the past tense

In Figure 1, does the RE refer to the RE of the WWTPs or the RE of different stages? This needs to be specified.

Line 94,what do PCCPs mean?

Line 102, the formulation of “nurdles/granules” is a bit unclear.

In Figure 2, note the formatting of the fonts in the figure.

Line 141-142, authors need to change the formulation of this sentence.

Line 142-143, references are needed about reagents.

In Table 1, authors can add a column about places. It might be better for authors to put references in the last column.

Line 173-183 and line 192-196, It's all reference 37, and this idea is meant to highlight what? It is recommended that this part be carefully conceived.

Line 217-219, it would be better to add the RE of MPs of different structures.

Line 420, this is all the content of the 47th reference, what is the reason for the other paragraph? The content can be more streamlined.

In “3.Filtration as a route of elimination”, authors can add some summary content or use tables to summarize the data that appear in these references.

In “3. Filtration as a route of elimination” and “4. Differences among MPs being filtrated”, some of the content is repetitive, such as the size of MPs. It might be better for the author to recheck it.

Reviewer 3 Report

The main contribution of this work is to summarize the background、source and treatment methods for micro-plastic. The paper is very well articulated, data are well described and reported.  Very well done. I have some concerns.

  1. The Introductionpart is too complicated and needs to be condensed. Authors are highly encouraged to include the point-by-point highlights of this article.
  2. The description form of the article needs to be adjusted! Please please do not just talk about one researches in one paragraph. To condense the research of multiple researches.

Round 2

Reviewer 1 Report

The submitted work can be accepted as it is. 

Reviewer 2 Report

The problems have been addressed.

Reviewer 3 Report

The revised paper addressed all my concerns, now I think it can be accepted by Energies now.